# Distinct epigenetic landscapes underlie the pathobiology of pancreatic cancer subtypes

Gwen Lomberk [1], Yuna Blum [2], Rémy Nicolle [2], Asha Nair[3], Krutika Satish Gaonkar[3], Laetitia Marisa [2], Angela Mathison[4], Zhifu Sun [3], Huihuang Yan [3], Nabila Elarouci[2], Lucile Armenoult[2], Mira Ayadi [2], Tamas Ordog [5,6], Jeong-Heon Lee[5], Gavin Oliver[3], Eric Klee[3], Vincent Moutardier[7,8], Odile Gayet[7], Benjamin Bian [7], Pauline Duconseil[7], Marine Gilabert[7], Martin Bigonnet[7], Stephane Garcia[7,8], Olivier Turrini[7,9], Jean-Robert Delpero[9], Marc Giovannini[9], Philippe Grandval[10], Mohamed Gasmi[8], Veronique Secq[8], Aurélien De Reyniès[2], Nelson Dusetti[7], Juan Iovanna[7] & Raul Urrutia [1,4,5]

Recent studies have offered ample insight into genome-wide expression patterns to define pancreatic ductal adenocarcinoma (PDAC) subtypes, although there remains a lack of knowledge regarding the underlying epigenomics of PDAC. Here we perform multi-parametric integrative analyses of chromatin immunoprecipitation-sequencing (ChIP-seq) on multiple histone modifications, RNA-sequencing (RNA-seq), and DNA methylation to define epigenomic landscapes for PDAC subtypes, which can predict their relative aggressiveness and survival. Moreover, we describe the state of promoters, enhancers, super-enhancers, euchromatic, and heterochromatic regions for each subtype. Further analyses indicate that the distinct epigenomic landscapes are regulated by different membrane-to-nucleus path-ways. Inactivation of a basal-specific super-enhancer associated pathway reveals the existence of plasticity between subtypes. Thus, our study provides new insight into the epigenetic landscapes associated with the heterogeneity of PDAC, thereby increasing our mechanistic understanding of this disease, as well as offering potential new markers and therapeutic targets.

[1] Division of Research, Department of Surgery, Medical College of Wisconsin, 8701 Watertown Plank Road, Milwaukee, Wisconsin 53226, USA.
[2] Programme Cartes d'Identité des Tumeurs (CIT), Ligue Nationale Contre Le Cancer, 14 rue Corvisart, Paris 75013, France. [3] Division of Biomedical Statistics and Informatics, Department of Health Science Research, Mayo Clinic, 200 First Street SW, Rochester, Minnesota 55905, USA. [4] Genomic Sciences and Precision Medicine Center, Medical College of Wisconsin, 8701 Watertown Plank Road, Milwaukee, Wisconsin 53226, USA. [5] Epigenomics Program, Center for Individualized Medicine, Mayo Clinic, 200 First Street SW, Rochester, Minnesota 55905, USA. [6] Department of Physiology and Biomedical Engineering, Mayo Clinic, 200 First Street SW, Rochester, MN 55905, USA. [7] Centre de Recherche en Cancérologie de Marseille (CRCM), INSERM U1068, CNRS UMR 7258, Aix-Marseille Université and Institut Paoli-Calmettes, Parc Scientifique et Technologique de Luminy, 163 Avenue de Luminy, Marseille 13288, France. [8] Hôpital Nord, Chemin des Bourrely, Marseille 13015, France. [9] Institut Paoli-Calmettes, 232 Boulevard Sainte Marguerite, Marseille 13009, France. [10] Hôpital de la Timone, 264 rue Saint-Pierre, Marseille 13385, France. These authors contributed equally: Gwen Lomberk, Yuna Blum, Rémy Nicolle. Correspondence and requests for materials should be addressed to G.L. (email: glomberk@mcw.edu) or to J.I. (email: juan.iovanna@inserm.fr) or to R.U. (email: rurrutia@mcw.edu)

Pancreatic ductal adenocarcinoma (PDAC) is a painful and fatal disease that undoubtedly remains a health priority, offers significant therapeutic challenges, and will soon rank as the second cause of death by cancer in the world[1]. Searching for somatic genetic causes, many laboratories have discovered oncogenes and tumor suppressors for PDAC[2]. However, cumulative evidence reveals more complex mechanisms underlying the development and progression of this disease, involving, among others, interactions between genomic and epigenomic alterations[3]. Although extensive studies have provided an understanding of aberrant gene networks[4], insights into the epigenomics of PDAC remains remarkably limited.

Epigenomics, which is the basis for the regulation of gene activity, expression, as well as nuclear organization and function includes the posttranslational modifications of histone proteins and DNA methylation. Histones contained within the basic repeating unit of chromatin, a nucleosome, can be dynamically modified at specific residues to signal for the activation or repression of transcription. Several modifications have been associated with particular transcriptional regulatory outcomes, including H3K4me3 with active promoters, H3K27ac with active enhancers and promoters, H3K4me1 with active and poised enhancers, H3K9me3 with heterochromatin, and H3K27me3 with Polycomb-repressed regions[5]. In addition, the combined distribution of different histone modifications reveals different epigenetic signals that mediate transcriptional initiation, elongation, and splicing, as well as DNA repair and replication, among others[6].

Here we have performed a multi-factorial integrative analysis of genome-wide chromatin immunoprecipitation-sequencing (ChIP-seq) on multiple histone modifications, as well as RNA-sequencing (RNA-seq) and DNA methylation studies to generate, for the first time, new knowledge on epigenetic landscapes linked to the heterogeneity of PDAC grown as patient-derived tumor xenografts (PDTXs). We report that PDTXs recapitulate two phenotypes observed in vivo, namely the classical and basal subtypes. Multi-parametric integrative analyses of these multi-omics datasets identify key epigenomic landscapes that are congruent with disease aggressiveness and survival. Super-enhancer mapping combined with transcription factor (TF) binding motif and upstream regulatory analyses reveal that these tumors populate two distinct epigenomic landscapes with classical tumors associated with TFs involved in pancreatic development, as well as metabolic regulators and Ras signaling, whereas the basal phenotype tumors utilize proliferative and epithelial-to-mesenchymal transition (EMT) transcriptional nodes downstream of the MET oncogene. The functional importance of these findings is underscored by the fact that genetic inactivation of MET results in a transition from a basal to more classical transcriptomic signature. Combined, this new knowledge on the PDAC epigenome, along with gene expression networks that it regulates, provides valuable and biomedically relevant mechanistic insight into this disease, offers potential new markers for PDAC, and informs the potential development of future therapeutic regimens that may help manage patients affected by this dismal malignancy. Therefore, these findings bear significant mechanistic and medical importance.

## Results

**Distinct chromatin states underlie PDAC heterogeneity.** PDTXs have become a promising tool to generate diagnostic, prognostic, and therapeutic approaches, particularly for individualized medicine. Importantly, these xenografts often recapitulate the biology, pathobiology, and therapeutic response of the primary counterpart, even though they do not actively reproduce the same microenvironment[7]. However, an important finding of the current study is that implantation of patient-derived tumors into mice reproduces two molecularly distinct subtypes of pancreatic cancer, namely basal and classical, indicating that these phenotypes are primarily maintained by the epithelial cell component. We report the first known multi-factorial integrative analysis of genome-wide ChIP-Seq for five distinct histone marks (H3K4me1, H3K27ac, H3K4me3, H3K27me3, and H3K9me3), DNA methylation, and RNA-seq on 24 human PDAC samples grown as PDTXs (Supplementary Table 1). Using a multivariate Hidden Markov Model, as built by ChromHMM[8], our analysis assigned fifteen chromatin states along with the average genome coverage by each state (Fig. 1a–c). In agreement with the ENCODE Roadmap project, we defined that H3K4me3 containing states are mostly found in promoter regions at the transcription start site (TSS) or flanking the TSS, often combined with the presence of H3K27ac and H3K4me1 (Fig. 1a). H3K4me3 near the TSS was associated with highly expressed genes, as determined by RNA-seq (E1 to E4; Fig. 1d) and low DNA methylation levels (Fig. 1c), unless present in combination with H3K27me3 (E5 and E6), which in general is a mark associated to strong downregulation (E5, E6, E13, and E15; Fig. 1b). Overall, the epigenetic states in PDAC correlated with clear effects on expression of nearby genes, as defined by RNA-seq (Fig. 1b). In addition, DNA methylation levels show specific patterns in association with gene expression that are chromatin state-dependent (Fig. 1c). Active promoters (E1 to E4) are, as expected, significantly hypomethylated and strong repressive states (E12, E13, and E15) are highly methylated. However, DNA methylation has a complex state-dependent role in gene regulation as hypermethylation can be associated to over-expression (e.g., E10 and E12) and hypomethylation to under-expression (e.g., E5 and E6). On average, 28% of each PDAC sample's epigenome was associated with silencing heterochromatin or repressed polycomb-based modifications (E12, E13, and E15; Fig. 1a). Approximately 57% of each epigenome had none of the tested histone marks, which, similar to reports in most cell types[9], results in a majority of genome categorized in the so-called quiescent state (E14). The remaining 15% of the epigenome corresponded to promoters and enhancers, in either an active or poised/bivalent state.

In order to characterize the impact of each of these specific chromatin states on pancreatic carcinogenesis, we identified cellular functions that were most frequently associated to each of these states (Fig. 1d, Supplementary Figure 1, and Supplementary Data 1). We found that virtually all aspects of pancreatic cancer biology were regulated by specific combinations of epigenetic marks and DNA methylation including the following: proliferation and apoptotic control (RB and TP53, other cell cycle checkpoints), major signaling pathways (ErbB, IGF, RAS, mammalian target of rapamycin, and transforming growth factor (TGF), among others), and cell adhesion molecules (such as cadherins and integrins). More specifically, we focused on the regulation of major pancreatic cancer genes and evaluated their tight transcriptional control by epigenetic modification (Fig. 1e). In particular, the major reprogramming TFs, *KLF4*, and *SOX9* (Fig. 1f and Supplementary Figure 2), as well as the *ERBB2* oncogene, were associated to marks of highly active TSS and active enhancers, as well as notable hypomethylation. On the other hand, the hedgehog pathway appeared strongly inhibited by the presence of extensive repressed-polycomb marks over the promoters and gene bodies of *SMO* and *PTCH1* (Fig. 1g and Supplementary Figure 2). Other tumor suppressor genes, such as *WT1*, *SMAD4*, and *BRCA2*, which are also silenced in all samples, are frequently associated to polycomb-repressed (E15, H3K27me3) or heterochromatin-like (E12, H3K9me3) states (Fig. 1e). Key epigenetic regulators were found to be significantly

upregulated by highly activated epigenetic states, potentially explaining this extensive epigenetic control of cancer-related genes. For instance, *DNMT1* (DNA methyltransferase), the repressing *EZH2* (H3K27 methyltransferase), and *HDAC1* (deacetylase), as well as the activating *MLL2*, *SETD3* (both H3K4 methyltransferases), and *KAT2A* (H3K acetyltransferase) are significantly overexpressed in association with active TSS and active enhancer chromatin in all samples (Fig. 1e).

Thus, this data indicates that the PDAC epigenome is dynamically marked for regulation, which together likely constitute areas that could react to extracellular signals, intrinsic cellular clues, and potentially inheritable modifications. The patterns of epigenetic marks on genes from major carcinogenic pathways demonstrate the dominant role of the epigenetic landscape on maintaining a neoplastic phenotype. As these marks are reversible, this data suggests epigenetic drugs

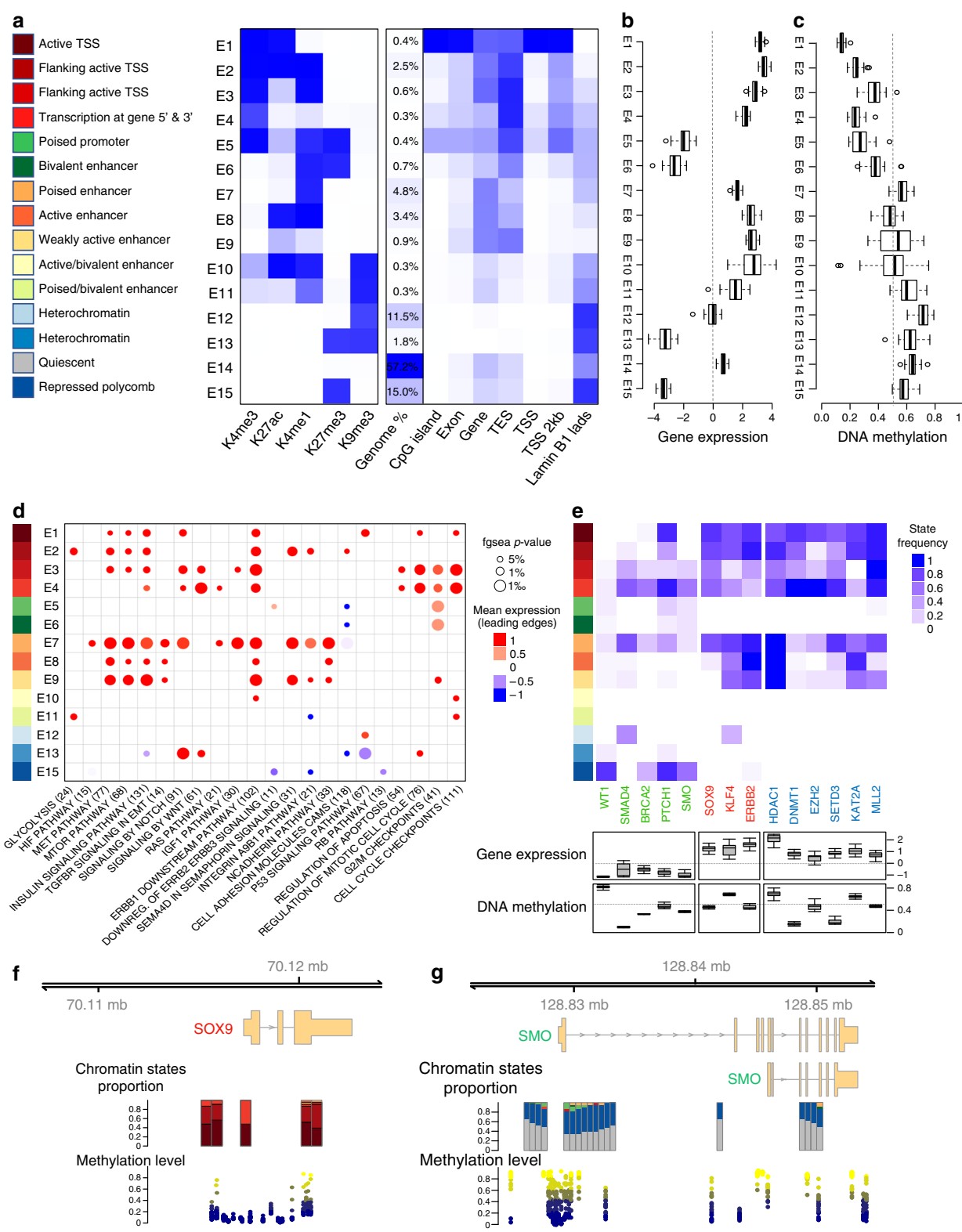

bear therapeutic potential, in particular for inhibitors of the highly upregulated molecules, EZH2 (e.g., tazemetostat), DNMT1 (e.g., decitabine), and HDAC1 (e.g., vorinostat and trichostatin A).

**Epigenomic landscapes differentiate clinical outcomes.** Subsequently, we sought to identify the chromatin states that can serve to epigenetically classify PDAC subtypes, by performing a MCA (multiple correspondence analysis) factorial analysis[10]. This method, which represents the categorical data generated by ChromHMM as points in a low-dimensional Euclidean space, behaves as the counterpart of the PCA (principal components analysis) that is commonly used for continuous data, allowing us to separate the tumors by their proximity. We found that the first MCA component was associated to a global increase of histone marks within regions gained in the genome, as determined by SNP (single-nucleotide polymorphism) array analysis (Supplementary Figure 3). This observation is extremely important, as it suggests that ChIP-seq comparisons, which do not employ this method to identify this type of epigenomic information in tumors with a myriad of duplications and deletions, may introduce technical bias yielding an increased probability of overestimating or underestimating histone marks in genomic regions with variable representation[11]. The second component was not influenced by any of these factors, thus offering the richest unbiased information for reliably classifying the tumors, as well as identifying gene networks of pathobiological importance. The third MCA component was mainly determined by one sample, which was hereafter considered as an outlier (Supplementary Figure 3). The outlier sample was a very differentiated tumor from a 60-year-old woman, which, surprisingly, was wild-type for *KRAS*, *SMAD4*, *CDKN2A*, and *p53* alleles, and displayed expression profiles and methylation patterns that were not characteristic of previously described pancreatic cancer tumors, suggesting that it may have been either clinically misclassified or represented a very rare type of pancreatic cancer not previously described. This interpretation further validated our classification method for the purpose of individualized medicine, which most often deals with single rare cases of diseases. Therefore, using the epigenomic regions associated to the second MCA component (5412 regions), the remaining 23 PDTXs were hierarchically clustered into two subtypes (Fig. 2a). Transcriptome- and methylation-based unsupervised analyses also supported this classification, thereby revealing the impact of both DNA and histone-based epigenetic components on gene expression (Fig. 2b and Supplementary Figure 4). Control analyses of the average levels of each histone modification in the enriched regions of the genome demonstrate that our classifications were not influenced by differences in overall levels of histone marks (Supplementary Figure 5). Cross-referencing our epigenomic data with published PDAC classifications based on genomic data[12, 13] clearly showed that these

subtypes of PDTXs correspond to the previously described classical and basal subtypes (Fig. 2c). These observations, plus additional support from a previous study on PDTXs[14], substantiate that these avatars retain features of human primary tumors and validate our analytical methodologies. Notably, although we found that the most frequent genomic alteration, including point mutations and copy number aberrations, do not distinguish subtypes (Supplementary Figure 6), the basal samples were more frequently associated to advanced, unresectable PDAC with liver metastasis (Fig. 2d). Overall, the integration of distinct chromatin states characterized the epigenetic landscape of PDAC to distinguish the less aggressive classical subtype from the more aggressive basal subtype. Further support of this observation was illustrated by the strong association of the second MCA component with patient survival (Fig. 2e). In summary, the use of multivariate histone-based information to develop a fifteen-state chromatin landscape model by ChromHMM followed by an MCA approach led to the classification of PDAC PDTXs into two major subgroups, which correlated with clinical parameters. Thus, this integrative method achieves better clinical value than each of the assays used independently, but more importantly it provides mapping of genome-wide epigenetic modifications, which can potentially serve as candidates for phenotypic, diagnostic, prognostic markers, and potential pharmacological targets from the significant growing number of epigenomic inhibitor drugs.

**Epigenomic landscapes implicate distinct pathways.** To gain biological and pathobiological mechanistic information on these PDAC subtypes, we performed unsupervised cluster analyses of the chromatin states defining the second MCA component assembled into three clusters of loci with particular epigenetic states, namely: cluster 1 mainly composed of enhancers active in the most basal-like samples; cluster 2 consisting of enhancers active in classical samples; and cluster 3 representing active promoters in classical samples (Fig. 3a and Supplementary Figure 7). These three clusters of loci were associated with differential methylation patterns, specifically with hypermethylation near enhancers in the basal samples and active promoters of classical samples (Fig. 3b). Each of these clusters of epigenetic landscapes was strongly associated with a corresponding change in transcription levels of nearby genes (Fig. 3c). The transcriptional activity of nearby genes was consistent with the predominant type of chromatin state in each cluster of loci, as denoted by an overall pattern of gene upregulation near regions of active enhancers (in basal samples for cluster 1 and classical samples for cluster 2) or active promoters (cluster 3 with classical samples only). Functional analysis of basal enhanced genes (cluster 1) revealed that, consistent with the aggressiveness of these tumors, these genes were implicated in signal transduction pathways with strong oncogenic potential (e.g., ErbB/EGFR,

**Fig. 1** Distinct chromatin states of human PDAC PDTXs. **a** Chromatin state definitions and histone mark probabilities as determined by ChromHMM[8]. Average genome coverage. Genomic annotation enrichments for each chromatin state as calculated by ChromHMM. **b** Boxplots illustrate sample centered averaged gene expression of genes within regions of particular chromatin states based on RNA-seq data. **c** Boxplots depict sample-averaged level of DNA methylation for the overlapping CpGs with each chromatin state. Dotted line represents mean methylation cut-off (0.5). **d** Gene-set enrichment analysis (GSEA) pathways for each chromatin state. Circle size is proportional to $-\log_{10} p$-value (showing only $p$-value < 5%) and colors correspond to the mean normalized expression of the genes driving the enrichment, or leading-edge genes. **e** Heatmap of the frequency of each state among all samples for tumor suppressors (green labels), pro-tumorigenic genes (red labels), and epigenetic regulators (blue labels). Boxplots of gene expression and methylation level are shown for each gene. **f,g** Visualization of chromatin state proportions on all samples and methylation levels across the *SOX9* locus and the *SMO* locus. Stacked bars represent the proportion of each chromatin state at a given genomic position. Methylation level is represented for each sample, at each genomic position and colored by its value. For all boxplots (**b**,**c**,**e**), bottom and top of boxes are the first and third quartiles of the data, respectively, and whiskers represent the lowest (respectively highest) data point still within 1.5 interquartile range of the lower (respectively upper) quartile. Center line represents the median value

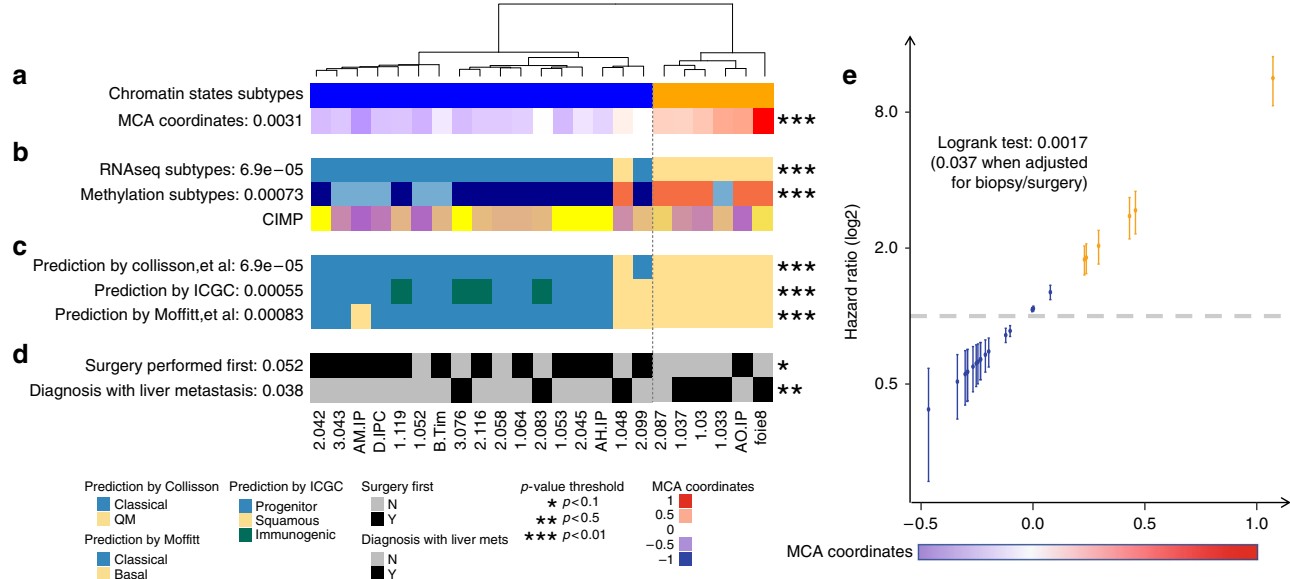

**Fig. 2** Epigenomic landscapes predict patient outcomes. **a** Chromatin state-based clustering using epigenetic regions associated with the second MCA component demonstrates two subtypes. Sample coordinates in this dimension and Student's *t*-test *p*-value of the association with chromatin state-based clustering are indicated. **b** Clustering of RNA-seq and DNA methylation data into subtypes is shown by distinct colors and plotted according to chromatin state-based clustering from **a**. *p*-values of Fisher's exact test demonstrate the association with chromatin state-based clustering. CIMP, CpG Island Methylator Phenotype calculated as the mean of island-CpG methylation level. **c** Prediction of PDAC subtypes using transcriptome-based publically available signatures[12, 13]. **d** Clinical characteristics of PDAC samples based on status of surgical resection and presence of liver metastasis at diagnosis and *p*-values of Fisher's exact test are indicated to show the association of clinical data with the chromatin state-based clustering. **e** Hazard Ratio (log2) estimated by a Cox model to demonstrate risk of death is shown along the MCA dimension. Dots are colored according to the chromatin state-based sample subtypes as defined in **a**

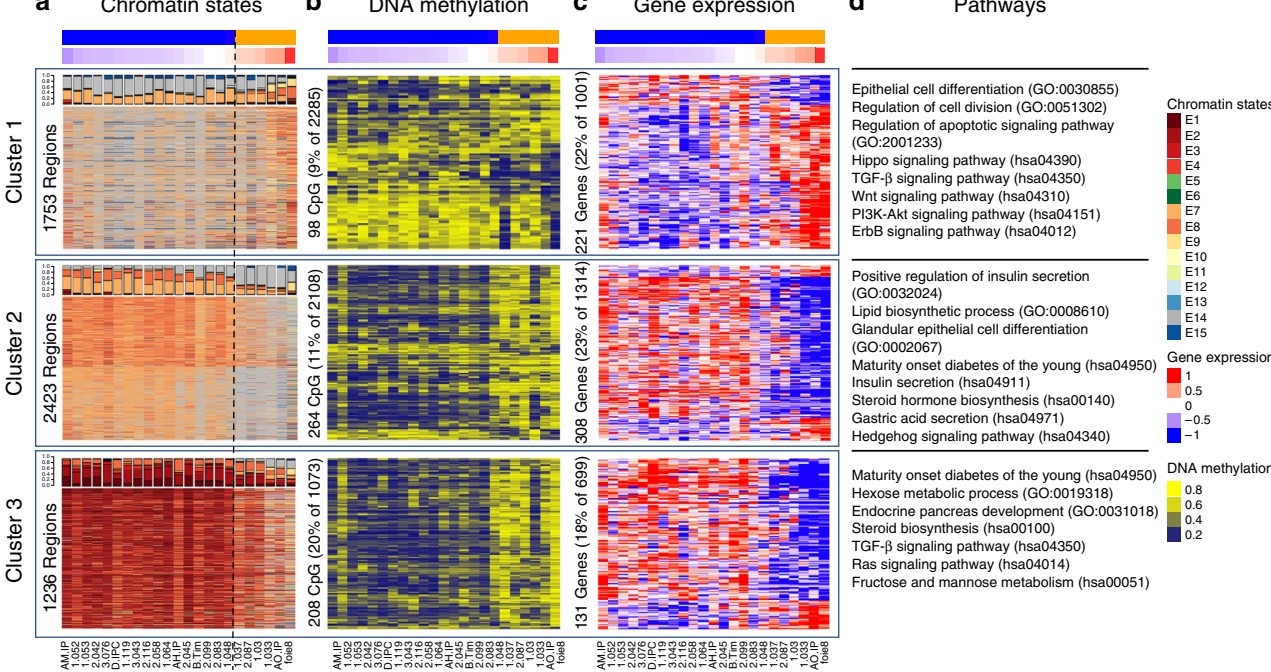

**Fig. 3** Epigenomic landscapes suggest pathobiological mechanisms. **a** Heatmap representing chromatin states for the regions associated with the second MCA dimension has been divided into three clusters of loci with differential epigenomic landscapes. Bar plots indicating proportion of chromatin states per sample are shown on the top of each cluster, based on the color code provided in Fig. 1a. **b** Blue-yellow heatmap of DNA methylation for nearby CpGs located < 1 kb upstream from regions shown in relation to the MCA coordinates (mad > 0.2 and *p*-value < 0.05, ANOVA test). **c** Blue-red heatmap of gene expression for nearby genes (mainly active TSS and mainly active enhancer regions located at < 20 kb and < 100 kb upstream from TSS, respectively) shown in relation to the MCA coordinates (mad > 0.2 and *p*-value < 0.05, ANOVA Test). **d** Pathways found to be significantly altered (*p*-value < 0.05, Fisher's exact test) in each cluster as determined by specific epigenomic landscapes. Pathway definitions originate from the GO or KEGG database

PI3K-AKT, Hippo, and Wnt), EMT, such as the TGFβ pathway, as well as deregulation of cell differentiation, proliferation and apoptosis (e.g., *YAP1*, *HEY1*, *MYC*, and *E2F7*) (Fig. 3d and Supplementary Table 2). Genes activated by epigenetic landscapes in classical samples (clusters 2 and 3) were mainly involved in pancreatic development (e.g., *PDX1*, *BMP2*, *GATA6*, *SHH*), metabolic processes (e.g., *HKDC1*, *FBP1*), and Ras signaling (e.g., *KITLG*, *RASA3*) (Fig. 3d and Supplementary Table 2). Thus, these results provide a better understanding of epigenetic landscapes that regulate biological pathways differentially enriched in each subtype of tumor, which likely explain the heterogeneity of PDAC.

**Super-enhancers reveal regulatory TF nodes for PDAC subtypes.** Additional mechanistic information was derived from characterizing the heterogeneity in the representation of super-enhancers in the different PDAC subtypes. Super-enhancers are known to have a cell- and state-specific function, as well as mediate the aberrant upregulation of cell fate determination genes in both developing and cancer cells[15]. We found the majority of super-enhancers to be specific for classical samples (250), including a substantial number of loci (28) associated with TFs, whereas only a few super-enhancers (30) were specific to basal samples. To further understand the regulatory programs driving the two phenotypes, we analyzed the upstream transcriptional regulation of these super-enhancers (Fig. 4). We found that the classical phenotype is likely influenced by at least 9 TFs contained within super-enhancers (*GATA6*, *FOS*, *FOXP1*, *FOXP4*, *KLF4*, *ELF3*, *NFIX*, *CUX1*, and *SSBP3*) (Fig. 4a and Supplementary Figure 8). In addition, these super-enhancer-associated TFs appear to exert their regulatory influence on other upregulated

TFs mostly associated with development, including TFs known to influence pancreatic morphogenesis (e.g., *HNF*s, *PDX1*, *MNX1*) and lipid metabolism (*PPAR*s). This information suggests the existence of a mechanism whereby enhancer-associated TFs amplify their function through regulating other fate-determining TFs and ultimately their target genes involved in distinct functions, in particular, metabolic networks for the classical subtype (Fig. 4a). Although no basal-specific TF was identified as an upstream regulator, *MET*, the hepatocyte growth factor (HGF) receptor, was associated with the regulation of basal-specific super-enhancers (Fig. 4b). This finding is important since anti-MET therapy is clinically used in other cancers[16]. Looking at the gene networks downstream of MET, we found that TFs primarily involved in proliferation, including *MYC*, *MYBL1*, and *E2F1*, and in EMT, such as *SNAI2*, are the best candidates to be key regulators of the basal phenotype.

When considering established knowledge on PDAC genetics and now, through the current study, the epigenetic landscape of these tumors, the data is in agreement with our previously described model for PDAC[3]. This model, refined by the data derived here, predicts that key epigenetic pathways, most likely working as effectors of well-known genetic alterations, serve as amplifiers and differentiating nodes to give rise to distinct PDAC phenotypes (Fig. 5a). It is likely that, at some moment of tumorigenesis, patient, environmental, and tumor-intrinsic factors (e.g., tumor microenvironment), or their combination push cells through various epigenetic landscapes. The spontaneous inter-conversion between subtype landscapes, once their phenotype is established, is unlikely. However, we believe that this could potentially be achieved through the inhibition of key pathways or processes, such as super-enhancers, which are predicted to be

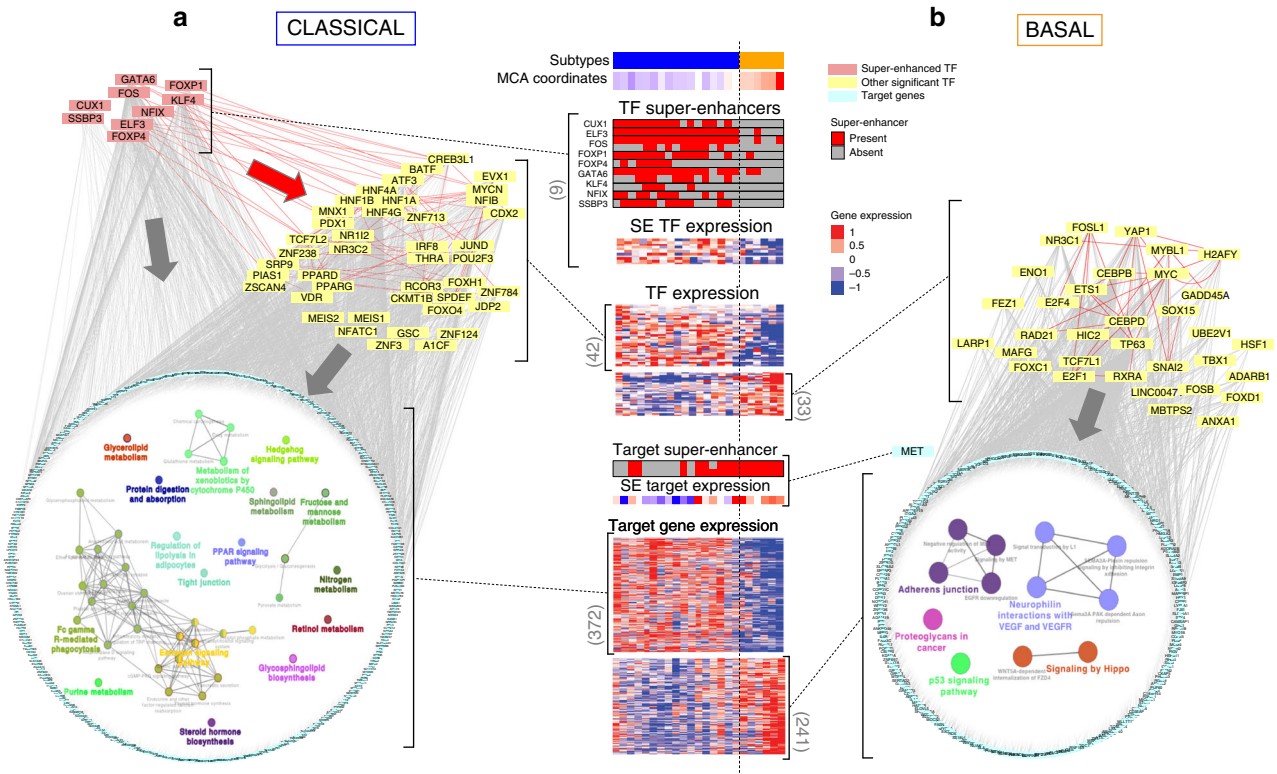

**Fig. 4** Transcriptional regulatory networks for PDAC phenotypes. TFs found to be significant regulators of the **a** classical- and **b** basal-associated genes were used to reconstruct regulatory networks. Networks include TFs upregulated by super-enhancers (upper-left light red nodes in **a** only), significant TFs (yellow), and their targets arranged in a circular layout. In each circle, cellular pathways and functions over-represented among the target genes of each network (classical **a**, basal **b**) are shown and grouped by gene-set similarity. In the center of **a** and **b**, the heatmaps of TF expression and their targets, as well as the heatmaps of super-enhancers are represented. Red arrow corresponds to TF–TF regulation, and gray arrows to TF–nonTF target regulations

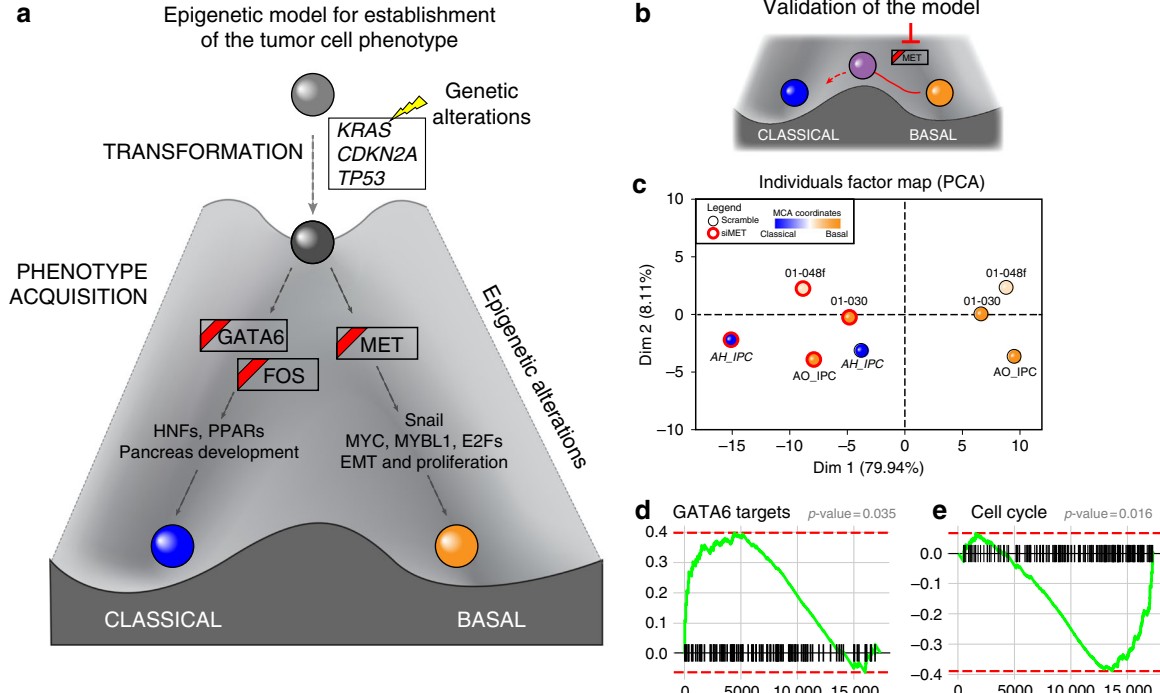

**Fig. 5** Epigenetic model for the tumor phenotype and its validation. **a** Genes that are in a box with red ribbon correspond to genes having an associated Super-enhancer. Yellow flash corresponds to genetic alterations. **b** Diagram is shown of the design for validating the model through inhibition of *MET* by siRNA to test the potential conversion of a basal to classical cellular phenotype. **c** PCA based on the differentially expressed genes from RNA-seq on basal siMET and basal control (scramble siRNA) samples. Basal samples were used as active individuals in the PCA construction, whereas the pure classical sample was projected on the two first dimensions. **d** GSEA plot is shown for GATA6 targets with vertical black lines corresponding to GATA6 putative targets ordered by their statistical tests of the differential analysis between siMET vs. control in basal samples. The curve illustrates the running enrichment score for the gene set ranked by the difference between the cohorts, showing global upregulation of GATA6 targets in the siMET condition. **e** GSEA plot for cell cycle is shown with vertical black lines corresponding to cell cycle genes (mitotic cell cycle, Reactome) ordered by their statistical tests of the differential analysis comparing siMET vs. control in basal samples. Green curve represents the running enrichment score for this gene set across the ranked list, which demonstrates global downregulation of those genes in the siMET condition

important for the acquisition of a distinct phenotype. Consequently, we tested the importance of MET in the maintenance of the basal phenotype, using small interfering RNA (siRNA) knockdown in PDTX-derived cell lines (Fig. 5b). Congruent with the epigenetic data, MET-inhibited samples underwent an overall shift towards the classical phenotype (Fig. 5c), which involved the increase of GATA6 transcriptional activity, as evidenced by upregulation of its gene targets (Fig. 5d), and the inhibition of cell cycle related pathways (Fig. 5e). Over-expression of apoptotic-related genes and metabolic modifications was also caused by manipulation of the MET pathway (Supplementary Figure 9 and Supplementary Data 2). Thus, these results reveal that subtypes have a potential plasticity, which can be manipulated by targeting pathways described in this study.

## Discussion
Epigenomic mechanisms are responsible for the regulation of ontologically related gene expression networks at an appropriate level, time, and place to give rise to both normal and diseased phenotypes. The current study provides the most comprehensive understanding, to date, of epigenomic landscapes underlying PDAC heterogeneity, predicts survival, informs the molecular pathobiology of this disease, as well as identifies epigenetically modified regions of the genome, which can serve as potential new markers and therapeutic targets. The integration of all datasets provides new information as to the state of promoter, enhancer, and super-enhancer associated transcriptional processes that are operational in the basal and classical PDAC phenotypes. When

modeled, our data leads to the inference that genetic, environmental, and tumor-intrinsic factors, such as the tumor microenvironment, likely all converge to give rise to distinct epigenetic landscapes during carcinogenesis. The inter-conversion between landscapes may not occur spontaneously, thereby fixing distinct subtypes. On the other hand, the targeting of upstream regulator pathways of key super-enhancers (e.g., MET), provides a proof-of-principle for the existence of phenotypic plasticity. Thus, we envision that through the guidance of this work, some agents may be useful for achieving this result. For instance, our results from targeting super-enhancer-mediated mechanisms of the more aggressive basal subtype, through MET inactivation, support the future testing of anti-MET therapies, such as those in clinical trials for many types of tumors[16], in PDAC. Furthermore, the data provided on both DNA methylation and histone marks support the testing of novel therapies with many drugs against writers, readers, and erasers that are in several phases of clinical trials or approved for other diseases[17]. Thus, we speculate that, in the future, pharmacological manipulation targeting specific pathways described here may convert the most aggressive tumors into a more benign or manageable counterpart in the clinic to improve survival.

## Methods
**Patient-derived xenografts**. Three expert clinical centers collaborated on this project after receiving ethics review board approval. Patients were included in this project under the Paoli-Calmettes Institute clinical trial number 2011-A01439-32. Consent forms of informed patients were collected and registered in a central database. The tumor tissues used for xenograft generation were deemed excess to

that required for the patient's diagnosis. PDAC tissue from surgical samples was fragmented, mixed with 100 μL of Matrigel, and implanted with a trocar (10 gauge; Innovative Research of America, Sarasota, FL) in the subcutaneous right upper flank of an anesthetized and disinfected male NMRI (Naval Medical Research Institute)-nude mouse. Samples obtained from EUS-FNA were mixed with 100 μL of Matrigel (BD Biosciences, Franklin Lakes, NJ) and injected in the upper right flank of a male nude mice (Swiss Nude Mouse Crl: NU(lco)-Foxn1nu; Charles River Laboratories, Wilmington, MA) for the first implantation. When xenografts reached 1 cm³, these were removed and passed to NMRI-nude mice in the same manner as the surgical samples. All animal experiments were conducted in accordance with institutional guidelines and were approved by the "Plateforme de Stabulation et d'Expérimentation Animale" (PSEA, Scientific Park of Luminy, Marseille).

**DNA and RNA extraction**. Nucleic acids were extracted for 24 xenograft samples corresponding to 24 unique patients. DNA was extracted using Blood & Cell culture DNA mini kit (Qiagen) following the manufacturer's instructions. RNA was extracted using RNeasy mini kit with optional on-column DNA digestion (Qiagen).

**Histone modification profiling (ChIP-seq) and analysis**. Tissue (50 mg) was homogenized for 15–30 s in 500 μL of 1 × PBS using a tissue grinder. Homogenized tissues were cross-linked to final 1% formaldehyde for 10 min, followed by quenching with 125 mM glycine for 5 min at room temperature, and by washing with tris-buffered saline (TBS). The pellets were resuspended in cell lysis buffer (10 mM Tris-HCl, pH 7.5, 10 mM NaCl, 0.5% NP-40) and incubated on ice for 10 min. The lysates were aliquoted into two tubes and washed with MNase digestion buffer (20 mM Tris-HCl, pH 7.5, 15 mM NaCl, 60 mM KCl, 1 mM CaCl₂) once. After resuspending in 250 μL of the MNase digestion buffer with proteinase inhibitor cocktails for each tube, the lysates were incubated in the presence of 1000 gel units of MNase (NEB, M0247S) per 4e6 cells at 37 °C for 20 min with continuous mixing in a thermal mixer. After adding an equal volume of sonication buffer (100 mM Tris-HCl, pH 8.1, 20 mM EDTA, 200 mM NaCl, 2% Triton X-100, 0.2% sodium deoxycholate), the lysates were sonicated for 15 min (30 s on; 30 s off) in a Diagenode bioruptor and centrifuged at 15,000 r.p.m. for 10 min. The cleared supernatant was incubated with 2 μg of histone modification-specific antibodies overnight at 4 °C. The following antibodies were used: anti-H3K4me1 (Abcam, ab8895, lot 114262), anti-H3K27ac, (Abcam, ab4729, lot GR150367), anti-H3K4me3 (Abcam, ab8580, lot GR188707-1), anti-H3K27me3 (Cell Signaling Technology, 9733 s, lot 8), and anti-H3K9me3 (Diagenode, C15410056, lot A1675-001p). After adding 30 μL of protein G-agarose magnetic beads, the reactions were incubated for another 3 h. Beads were washed extensively with ChIP buffer, high-salt buffer, LiCl₂ buffer, and TE buffer. Bound chromatin was eluted and reverse-crosslinked at 65 °C overnight. DNA was purified using the Mini-Elute PCR purification kit (Qiagen) after treatment with RNase A and proteinase K. Enrichment was confirmed by targeted real-time PCR in positive and negative genomic loci. For next-generation sequencing, ChIP-seq libraries were prepared from 10 ng of ChIP, and input DNA with the Ovation Ultralow DR Multiplex system (NuGEN). The ChIP-seq libraries were sequenced to 51 base pairs from both ends using the Illumina HiSeq 2000 in the Mayo Clinic Medical Genomics Core. Data were analyzed using the HiChIP pipeline[18]. Briefly, paired-end reads were mapped by BWA[19] and pairs with one or both ends uniquely mapped were retained. H3K4me3, H3K4me1, and H3K27ac peaks were called using the MACS2 software package[20] at false discovery rate (FDR) ≤ 1%. SICER[21] was used to identify enriched domains for H3K27me3 and H3K9me3. For data visualization, BEDTools[22] in combination with in-house scripts were used to generate normalized tag density profile at a window size of 200 bp and step size of 20 bp. We also visualized the average profile around TSS for H3K4me1, H3K4me3, H3K9me3, H3K27ac, and H3K27me3 across all samples. The tags were normalized to tags-per-million with a flanking region of 10 kb around the TSS. The data was plotted with a Y-axis as the normalized log2 fold change of IP over control and the X axis as the bins across the given region of interest. Unsupervised clustering was performed by selecting the top 20,000 merged peak regions with highest variance to generate the clusters for individual histone modifications.

**DNA methylation profiling and analysis**. Whole-genome DNA methylation was analyzed using the Illumina Infinium MethylationEPIC Beadchip. Integragen SA (Evry, France) carried out microarray experiments and hybridized to the BeadChip arrays following the manufacturer's instructions. Illumina GenomeStudio software was used to extract the probe DNA methylation intensity signal values for each locus. Data were then preprocessed following recommendations from the Dedeurwaerder et al[23]. Data were removed from probes that were not detected or saturated and that contained SNPs or overlapped with a repetitive element that was not uniquely aligned to the human genome or regions of insertions and deletions in the human genome. Data were then adjusted for color balance bias and normalized between samples using the SSN (shift and scaling normalization) method using the lumi package functions. The CpG Island Methylator Phenotype (CIMP) index was estimated by adapting the approach by Toyota et al[24]. In brief, CpG islands found to be unmethylated ( < 20% β-value) in all 25 normal pancreatic samples from the ICGC consortium were selected. The CIMP index was calculated independently for

each sample as the proportion of methylated ( > 30% β-value) CpGs among the selected normally unmethylated island CpG.

**mRNA profiling (RNA-seq) and analysis**. RNA libraries were prepared (Illumina TruSeq RNA v2) and run on the Illumina High Seq-2000 for 101 bp paired end reads in the Mayo Clinic Medical Genomics Core. Gene expression profiles were obtained using the MAP-RSeq v.1.2.1 workflow[25], the Mayo Bioinformatics Core pipeline. MAP-RSeq consists of alignment with TopHat 2.0.6[26] against the human hg19 genome build and gene counts with the HTSeq software 0.5.3p9 (http://www.huber.embl.de/users/anders/HTSeq/doc/overview.html) using gene annotation files obtained from Illumina (http://cufflinks.cbcb.umd.edu/igenomes.html). RNA-seq reads were also mapped using STAR[27] with the proposed ENCODE parameters and XENOME[28] on the human hg19 and mouse mm10 genomes and transcript annotation (Ensembl 75). Gene counts were normalized using reads per kilobase per million mapped reads (RPKMs).

**Public dataset comparison**. ICGC Methylation chips, RNAse, and microarray gene expression datasets were downloaded from the ICGC data portal (dcc.icgc.org, release 20). Other datasets were downloaded from the provided Gene Expression Omnibus entry (Moffitt et al. GSE71729[13] and Collisson et al.[12] GSE17891). All non-cancer samples were removed from each dataset. Expression datasets were then centered gene-wise. Centroid classifiers were built for each dataset describing a classification using an approach described in previous works[29, 30]. Briefly, after gene-wise centering, the 1000 most differentially expressed genes (limma) or differentially methylated CpG (Student's t-test), were used to build centroids of each subtype. Gene expression profiles of samples to test were correlated (Pearson's correlation) to all the centroids of a classification system and the closest centroid class (highest correlation coefficient) was assigned. When specified, indeterminate samples correspond to samples that did not significantly correlated to any centroid.

**SNP arrays analysis**. Illumina Infinium HumanCode-24 BeadChip SNP arrays were used to analyze the DNA samples. Integragen SA (Evry, France) carried out hybridization, according to the manufacturer's recommendations. The BeadStudio software (Illumina) was used to normalize raw fluorescent signals and to obtain log R ratio (LRR) and B allele frequency (BAF) values. Asymmetry in BAF signals due to bias between the two dyes used in Illumina assays was corrected using the tQN normalization procedure[31]. We used the circular binary segmentation algorithm[32] to segment genomic profiles and assign corresponding smoothed values of LRR and BAF. The Genome Alteration Print method was used to determine the ploidy of each sample, the level of contamination with normal cells, and the allele-specific copy number of each segment[33].

**Unsupervised clustering**. Features on sexual chromosomes were removed for subsequent analysis. One sample outlier identified by MCA analysis was removed for clustering. Unsupervised clustering analysis was carried out on: gene expression (RNA-seq, 23 samples), CpG methylation (MethEpic, 23 samples), and H3K4me3, H3K4me1, H3K27ac, H3K9me3, and H3K27me3 (ChIP-seq, 23 samples). For histone marks, ChIP-seq data were filtered as follows: − log10(FDR) > 10 and number of samples that share the peak > 3. For sequence or ChIP-based data, an extension of the ConsensusClusterPlus algorithm was used[34]. In brief, using all paired combination of Pearson's distance and different linkage metrics (Ward, complete and average), hierarchical clustering is bootstrapped in 1000 iterations of resampling of the most variant features. An additional level of iteration adjusts the threshold of feature variability. The consensus is given by a final hierarchical clustering using the complete linkage and the number of co-classification as sample distance. For Methylation chips, the SD was used as a measure of variability and 10 thresholds between 1% and 10% were used for each iteration of ConsensusClusterPlus. To consider the specificity of the mean-variance relationship in count data, a combination of mean and SD of the log-counts (minimum rank of both) was used to select between 1% and 50% of the features with the highest counts and variance in RNA-seq and ChIP-seq data. Chromatin state-based clustering was performed on the 5412 regions that were associated to the second MCA components using a binary distance and Ward linkage metrics. Clustering of chromatin states regions was performed using hierarchical clustering and the number of clusters was determined using the cutreeDynamic function (dynamicTreeCut R package[35]) with a minimum size module of 500 features.

**ChromHMM**. ChromHMM[8] was used to perform hidden Markov modeling on the five histone marks and, by default, chromatin states were analyzed at 200 bp intervals and a fold threshold of 10. The tool was used to learn consensus models from virtually concatenated aligned bam files from all the samples. Control data were used in the model to help reduce copy number variation and repeat associate artifacts in the ChIP-seq samples. Multiple state models were visualized to capture all the key chromatin states. A 15-state model was selected and applied on all samples to obtain overlap and enrichment for each state. Enrichment of each state was calculated across genomic regions of interest and visualized as heatmaps for genomic regulatory regions of interest. The states were then given functional annotation based on these enrichment patterns.

**MCA analysis**. ChromHMM output files were concatenated using the unionbed function from BEDTools[22]. Chromatin states (4,810,649 regions) were filtered so that the 2 main representative states per region were present in at least 30% of samples, and that the quiescent states (E14) was not the main representative state per region. Sex chromosomes were removed for the rest of the analysis. An MCA analysis[10] was performed on the filtered chromatin states matrix (665,328 regions) using FactoMineR[36]. As in PCA, the MCA decomposes the variance into a component of a set of new orthogonal variables (dimensions) ordered by the amount of variance that each component explains. For a given dimension, the most contributing regions were selected as those having a significant association (Anova test, $a$ 5%) and having a determination coefficient above 0.6 (dimdesc function from FactoMineR).

**Functional analysis**. In order to retrieve the pathways affected by a particular chromatin state, a gene-set enrichment analysis (GSEA) approach was performed for each chromatin state, using the fgsea R package[37], which implements GSEA on a pre-ranked genelist and MsigDB signaling database[38]. Gene score used in fgsea was the frequency among all samples of the presence of a particular chromatin state within $-20$ kb from TSS $+1$ kb from TSE of the gene. Leading edges correspond to genes that drive the enrichment as given by fgsea.

For the regions contributing the most to the second MCA dimension (5412 regions selected as described in the previous paragraph), we retrieved nearby CpGs and genes. Nearby CpGs were determined using the findOverlaps function from the IRanges packages[39] and with a location < 1 kb upstream from regions. Nearby genes were retrieved using rGreat[40] (basal plus extension; proximal 5 kb upstream 1 kb downstream, plus distal: up to 20 kb for Active TSS regions and 100 kb for active enhancer regions). Pearson's correlation ($a$ 5%) with the MCA component coordinates was used to select associated genes and CpGs. We performed functional enrichment analysis on the genes corresponding to each region clusters using the Enrichr software[41].

**TF and super-enhancer analysis**. ROSE[42, 43] was employed to identify super-enhancers based on their density and length of the stitched regions. A peak stitching distance of 12,500 bp was used and the regions around TSS 2500 bp were excluded, while identifying super-enhancers in each sample. The signal and ranks are normalized from 0 to 1 and sorted by their score and plotted. All stitched regions above the inflection point were considered as super-enhancers for further analysis. BAM tracks were visualized using the gviz R package[44]. Significant TFs were identified by gene-target enrichment analysis using the ChEA database (Fisher exact test on the basal-classical component associated genes)[45] or by motif enrichment on the basal-classical component associated regions using PWMEnrich R package. TFs, for which expression was negatively or positively correlated to the basal-classical component, were regarded as classical or basal TFs, respectively. Cytoscape[46] was used for network visualization and the ClueGO app[47] was applied to target genes for clustered enriched pathway network representation.

**Met siRNA transfection, RNA-seq, and qPCR validations**. Cells (3e5) were plated in six-well plates and 24 h later transfected with a pool of 4 *MET* siRNAs (ON-TARGETplus siRNA Reagents, Dharmacon), using INTERFERin reagent (Polyplus-transfection) according to the manufacturer's protocol. A scrambled siRNA pool was used as the negative control. After 72 h, cells were lysed, and RNA extracted with RNeasy Mini Kit (Qiagen). The sequences of Met-specific siRNAs were as follows: Met1: 5′AACUGGUGUCCCGGAUAU-3′; Met2: 5′-GAA-CAGCGAGCUAAAUAUA-3′; Met3: 5′-GAGCCAGCCUGAAUGAUGA-3′; and Met4: 5′-GUAAGUGCCCGAAGUGUAA-3′. RNA libraries were prepared (Illumina TruSeq RNA v2) and run on the Illumina High Seq-2500 for 125 bp paired end reads in the Genomic Sciences and Precision Medicine Center, Medical College of Wisconsin. Gene counts were normalized using RPKM. Student's $t$-tests were performed to test for differentially expressed genes (list is given in Supplementary Table 3) between the siMET basal and scramble basal samples. Fgsea enrichment tests were performed based on Pvalues of the differential analysis of basal siMET vs. basal control samples, and by using MsigDB database and the described open resource[48]. For qPCR, total RNA (1 µg) was used as a template for cDNA synthesis, using the GoScript™ reverse transcription kit (Promega). GoTaq® qPCR 2 × Master Mix (Promega) was used with the following reaction conditions: denaturation at 95 °C for 2 min; 40 cycles of 15 s at 95 °C, 45 s at 60 °C. Reactions were carried out using the AriaMx real-time PCR system and analyzed using the AriaMx software v1.1 (Agilent Technologies, Santa Clara, CA, USA). Primer lists for each transcript are provided in Supplementary Table 4.

**Data availability**. ChIP-seq, DNA methylation, RNA-seq, and SNP datasets that support the findings of this study have been deposited at ArrayExpress (http://www.ebi.ac.uk/arrayexpress) under accession codes E-MTAB-5632, E-MTAB-5571, E-MTAB-5639, and E-MTAB-5570, respectively.

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

## Acknowledgements

This work was supported by La Ligue Contre le Cancer (Programme Cartes d'Identité des Tumeurs® (CIT)), INCa, Canceropole PACA, DGOS (labellization SIRIC), INSERM to J.I., the National Institutes of Health (grants R01 CA178627 to G.L., and R01 DK52913 to R.U.).

## Author contributions

G.L., A.M., N.E., and J.H.L. made contributions to the acquisition of data. V.M., P.D., M.G., S.G., O.T., J.R.D., M.G., P.G., M.G., and V.S. contributed to sample acquisition. O.G., B.B., and M.B. established and maintained the PDTXs. G.L., Y.B., R.N., A.N., K.S.G., L.M., Z.S., H.Y., N.E., L.A., M.A., G.O., E.K., J.I., and R.U. contributed to analysis and interpretation of data. G.L., J.I., and R.U. generated the main idea of the work and developed the study design, both conceptually and methodologically. G.L., A.N., A.D.R., J.I., and R.U. supervised the data analysis. G.L., N.D., J.I., and R.U. supervised the project. G.L., Y.B., R.N., L.M., A.N., J.I., and R.U. wrote the manuscript from first draft to completion. K.S.G., A.M., N.E., L.A., M.A., T.O., J.H.L., G.O., A.D.R., and N.D. made comments, suggested appropriate modifications, and corrections. All authors read and approved the final manuscript.

## Additional information

**Competing interests:** The authors declare no competing interests.

