## [Peer Review File · Nature Communications]

Reviewers' comments:

Reviewer #1:

Manuscript Review:

Distinct epigenetic landscapes underlie the pathobiology of pancreatic cancer subtypes (Lomberk et al. 2017)

In this manuscript, Lomberk et al seek to undercover the epigenetic landscapes of pancreatic ductal adenocarcinoma (PDAC) through the use of patient-derived xenografts (PDXs). By employing an integrative approach to characterize global histone modifications, gene transcription and DNA methylation patterns, the authors were able to find key epigenetic states that associate with PDAC disease aggressiveness and patient survival, correlating with previously described transcriptional subtypes (classical and basal/quasi-mesenchymal). Additionally, through super-enhancer mapping and transcription factor motif analysis, Lomberk et al also found that the classical subtype of PDAC is associated with transcription factors involved in pancreatic development, metabolic regulation, as well as RAS-signaling. In contrast, there was a basal-specific super-enhancer associated with the MET oncogene proliferation in the basal subtype of PDAC. Taken together, the authors propose that understanding the epigenetic status in relation with gene expression, offers new insight into PDAC disease pathology and could inform future development of therapeutic regimes.

Overall comments:

The authors have invested a great deal of effort in performing, analyzing and interpreting the experiments, and the manuscript provides a useful resource in documenting epigenetic states in pancreatic cancer. However, the authors should more effectively link their epigenetic data with disease features and cancer-causing pathways. Secondly, the authors need to perform some follow up experiments to validate the predicted findings of aberrant disease pathways, super-enhancer or transcription factor activity

Specific comments:

1. In this manuscript, the authors performed analysis using 24 PDXs derived from PDAC patients. However, the authors have not provided detailed clinical-pathological information of these patients. This information is necessary and critical towards creating a robust groundwork to build upon subsequent findings. This should include age, sex and ethnicity, the AJCC pathology stage and grade, and tumor location (primary or metastatic region). A good example to follow is that from the study by Bailey et al (1). Whether the PDACs were associated with cystic precursors (IPMN or MCN) or have atypical histology should be noted.
2. No information about the key pancreatic cancer gene mutations is provided. Understanding the mutational profile for each PDX is important for understanding the relationships between epigenetic states, genomic changes, and disease subtypes.
3. The authors suggest that the presence of classical/basal signatures among the PDXs indicates that these models 'retain features of the original human tumors' (lines 157-159). However, there is no actual data to substantiate this (i.e. no RNA-seq data or ChIP-seq data are presented on corresponding primary tumor material). Overall, some degree of validation of the epigenetic data in corresponding primary tumors should be performed.
4. The mapping of epigenetic states and association with transcriptional programs is interesting. However, the authors do not provide any functional validation of the predicted circuits emerging from their analysis (e.g. such as could be performed using organoids derived from the PDXs).
 - Nine transcription factors were contained within super-enhancers in the classical PDAC subtype (Fig. 4a), and the authors proposed that these super-enhancer associated transcription factors regulate other downstream transcription circuits. These include pancreatic morphogenesis and lipid

metabolism. However, these claims were not experimentally verified and as such, remain as bioinformatic predictions. Will the CRISPR knockout of single or combinations of these 9 super-enhancer associated transcription factor result in attenuated growth of classical-subtypes of PDAC? What functional role does MET play in the basal tumors? Overall, one would like to see at least one or two pathways emerging from the analysis to be examined, such as the role of MET, or of specific transcription factors associated with the classical state (e.g. in control of expression signatures and/or growth)

References:

1.. Bailey P, Chang DK, Nones K, Johns AL, Patch AM, Gingras MC, et al. Genomic analyses identify molecular subtypes of pancreatic cancer. *Nature*. 2016 Mar 3;531(7592):47-52.

Reviewer #2:

The authors have carried out molecular profiling of a series of 24 xenografts from human pancreatic cancers, using RNA-Seq, CpG methylation Beadchips, SNP arrays for DNA copy number, and ChIP-seq for histone modifications. Their data, for each of the epigenetic and expression modalities, clusters "basal type" away from "classical type" PDAC xenografts.

Lines 227-259 in the manuscript summarize the key findings - namely that a specific group of transcription factors (TFs) are upregulated and epigenetically marked by active chromatin states in their enhancer regions in the "classical type" xenografts, while a different set of TFs show these characteristics in the "basal type" xenografts.

These findings are interesting and potentially useful, and the Figures summarizing the "shape of the data", in particular with appropriate bioinformatic enrichment analyses, are well designed and informative. Also, the correlations observed between CpG methylation, gene expression, and specific types of chromatin states are interesting and potentially broadly relevant.

However, as it stands, this study is purely correlative in design and is basically a thorough bioinformatic analysis of now fairly standard types of molecular information from a modestly sized series of PDAC xenografts. One feels that it needs at least one or two manipulative experiments for hypothesis testing - i.e. perturb the system (particularly by down-regulating the key TFs) and ask what happens both to cell proliferation and apoptosis and, importantly, to the putative gene networks that the authors have identified. Being able to do such experiments is, after all, a main purpose of making xenografts.

We appreciate the reviewers for their complimentary comments, including “*The authors have invested a great deal of effort in performing, analyzing and interpreting the experiments, and the manuscript provides a useful resource in documenting epigenetic states in pancreatic cancer*” (reviewer 1); and “*These findings are interesting and potentially useful, and the Figures summarizing the "shape of the data", in particular with appropriate bioinformatic enrichment analyses, are well designed and informative. Also, the correlations observed between CpG methylation, gene expression, and specific types of chromatin states are interesting and potentially broadly relevant*” (reviewer 2).

We are similarly grateful for their generous suggestions on how to improve this manuscript even further. Below, we provide a detailed response to all points raised by the reviewers.

DETAILED RESPONSE TO REVIEWERS:

REVIEWER 1:

Comment 1: In this manuscript, the authors performed analysis using 24 PDXs derived from PDAC patients. However, the authors have not provided detailed clinical-pathological information of these patients. This information is necessary and critical towards creating a robust groundwork to build upon subsequent findings. This should include age, sex and ethnicity, the AJCC pathology stage and grade, and tumor location (primary or metastatic region). A good example to follow is that from the study by Bailey et al (1). Whether the PDACs were associated with cystic precursors (IPMN or MCN) or have atypical histology should be noted.

Response 1: We thank the reviewer for this suggestion, which we now provide as Supplementary Table 1. Patients ranged in age from 46 to 81 years old (Av:64, SD: 9.4) and were almost equally divided in gender (F=11; M=13). Two different TNM stages dominated the progression score, namely 4 (45.5%) and 2b (50%), while one was 2a and 2 remained undetermined.

Importantly, no correlation of any of these clinical parameters influence the classification of the tumor or their epigenetic characteristics.

Comment 2: No information about the key pancreatic cancer gene mutations is provided. Understanding the mutational profile for each PDX is important for understanding the relationships between epigenetic states, genomic changes, and disease subtypes.

Response 2: In response to this important request from the reviewer, we have added Extended Data Figure 6 along with lines 161-162, showing the most frequent genomic alterations in 16 of our samples, including point mutations and copy number aberrations, which we find not to be discriminative of the tumor subtypes or a landscape.

Comment 3: The authors suggest that the presence of classical/basal signatures among the PDXs indicates that these models ‘retain features of the original human tumors’ (lines 157-159). However, there is no actual data to substantiate this (i.e. no RNA-seq data or ChIP-seq data are presented on corresponding primary tumor material). Overall, some degree of validation of the epigenetic data in corresponding primary tumors should be performed.

Response 3: We thank the reviewer for helping us clarify our previous statement, as we have only shown that the basal/classical methylation and mRNA signatures from PDX are similar to human

in situ tumors in general. To better reflect our observations, we have modified the manuscript as follows (lines 156-160).

Cross-referencing our epigenomic data with published PDAC classifications based on genomic data^{10,11} clearly showed that these subtypes of PDTXs correspond to the previously described classical and basal subtypes (Fig. 2c). These observations, plus additional support from a previous study on PDTXs¹², substantiate that these avatars retain features of human primary tumors and validate our analytical methodologies.

Comment 4: The mapping of epigenetic states and association with transcriptional programs is interesting. However, the authors do not provide any functional validation of the predicted circuits emerging from their analysis (e.g. such as could be performed using organoids derived from the PDXs).

- Nine transcription factors were contained within super-enhancers in the classical PDAC subtype (Fig. 4a), and the authors proposed that these super-enhancer associated transcription factors regulate other downstream transcription circuits. These include pancreatic morphogenesis and lipid metabolism. However, these claims were not experimentally verified and as such, remain as bioinformatic predictions. Will the CRISPR knockout of single or combinations of these 9 super-enhancer associated transcription factor result in attenuated growth of classical-subtypes of PDAC? What functional role does MET play in the basal tumors? Overall, one would like to see at least one or two pathways emerging from the analysis to be examined, such as the role of MET, or of specific transcription factors associated with the classical state (e.g. in control of expression signatures and/or growth)

Response 4: To address these concerns, we took cells derived from PDTXs of both subtypes, treated them with siRNA against C-Met and performed RNA-Seq analysis combined with confirmation by q-PCR. These new data have been included in the revised manuscript (lines 275-282) and as new Figure 5b-e, Extended Data Figure 9 and Supplementary Tables 4-6. Notably, as predicted by our model, the inactivation of C-Met favors the maintenance of a classical phenotype, which even the ability of basal cells to acquire the characteristics of this more favorable tumor subtype. Thus, these new experiments provide a proof-of-principle that the epigenetic landscapes, though apparently stable, can be converted by key manipulations. We are excited with these results since they not only validate our model, but also bear significant medical relevance for the future development of therapies against this disease.

REVIEWER 2:

Comment: The authors have carried out molecular profiling of a series of 24 xenografts from human pancreatic cancers, using RNA-Seq, CpG methylation Beadchips, SNP arrays for DNA copy number, and ChIP-seq for histone modifications. Their data, for each of the epigenetic and expression modalities, clusters "basal type" away from "classical type" PDAC xenografts.

Lines 227-259 in the manuscript summarize the key findings - namely that a specific group of transcription factors (TFs) are upregulated and epigenetically marked by active chromatin states in their enhancer regions in the "classical type" xenografts, while a different set of TFs show these characteristics in the "basal type" xenografts.

These findings are interesting and potentially useful, and the Figures summarizing the "shape of the data", in particular with appropriate bioinformatic enrichment analyses, are well designed and informative. Also,

the correlations observed between CpG methylation, gene expression, and specific types of chromatin states are interesting and potentially broadly relevant.

However, as it stands, this study is purely correlative in design and is basically a thorough bioinformatic analysis of now fairly standard types of molecular information from a modestly sized series of PDAC xenografts. One feels that it needs at least one or two manipulative experiments for hypothesis testing - i.e. perturb the system (particularly by down-regulating the key TFs) and ask what happens both to cell proliferation and apoptosis and, importantly, to the putative gene networks that the authors have identified. Being able to do such experiments is, after all, a main purpose of making xenografts.

Response: We agree with both reviewers' comments and thus, we have performed additional experiments that are described above for **Reviewer 1, Comment 4**.

REVIEWERS' COMMENTS:

Reviewer #1 (Remarks to the Author):

The authors have nicely revised their paper to clarify a number of points, to incorporate additional genomic and clinical information relating to the PDXs, and to include functional analysis of one candidate mediator for the basal-like subtype emerging from their chromatin studies (MET). This study represents a valuable contribution to the field.

Reviewer #2 (Remarks to the Author):

The authors of this molecular study of pancreatic adenocarcinoma PDX's have responded to the two critiques with new data. The data addressing my main request (functional analysis of the candidate pathways in at least one of the PDX's) are from an siRNA knockdown experiment, in which down-modulation of MET expression was found to correlate with global up-regulation of GATA6 targets, and down-regulation of mitotic cell cycle marker genes, in the siMET condition. This new experiment satisfies my request, and I believe the manuscript will now be well received and quite useful for this obviously important field of cancer research.